# Deep Learning-Based Subtask Segmentation of Timed Up-and-Go Test Using RGB-D Cameras

**DOI:** 10.3390/s22176323

**Published:** 2022-08-23

**Authors:** Yoonjeong Choi, Yoosung Bae, Baekdong Cha, Jeha Ryu

**Affiliations:** School of Integrated Technology, Gwangju Institute of Science and Technology (GIST), Gwangju 61005, Korea

**Keywords:** timed up-and-go test, TUG subtask segmentation, deep learning, temporal convolutional network

## Abstract

The timed up-and-go (TUG) test is an efficient way to evaluate an individual’s basic functional mobility, such as standing up, walking, turning around, and sitting back. The total completion time of the TUG test is a metric indicating an individual’s overall mobility. Moreover, the fine-grained consumption time of the individual subtasks in the TUG test may provide important clinical information, such as elapsed time and speed of each TUG subtask, which may not only assist professionals in clinical interventions but also distinguish the functional recovery of patients. To perform more accurate, efficient, robust, and objective tests, this paper proposes a novel deep learning-based subtask segmentation of the TUG test using a dilated temporal convolutional network with a single RGB-D camera. Evaluation with three different subject groups (healthy young, healthy adult, stroke patients) showed that the proposed method demonstrated better generality and achieved a significantly higher and more robust performance (healthy young = 95.458%, healthy adult = 94.525%, stroke = 93.578%) than the existing rule-based and artificial neural network-based subtask segmentation methods. Additionally, the results indicated that the input from the pelvis alone achieved the best accuracy among many other single inputs or combinations of inputs, which allows a real-time inference (approximately 15 Hz) in edge devices, such as smartphones.

## 1. Introduction

The timed-up-and-go (TUG) test is an effective tool for assessing an individual’s functional mobility that includes the most basic and essential activities in daily life, such as standing up, walking, turning, and sitting back [1,2]. Because the TUG test is relatively easy, it has been widely tested for individuals with balance and gait impairments, such as Parkinson’s disease (PD) [3], total knee arthroplasty (TKA) [4], stroke [5], multiple sclerosis (MS) [6], lumbar degenerative disc disease [7], lower limb amputations [8], chronic obstructive pulmonary disease (COPD) [9], and cognitive decline [10], to assess their risk of falling [2,3].

In a typical TUG test, an individual starts sitting on the chair, rises from the chair, walks a set distance of 3 m or 5 m, turns around, returns to the chair, and finally sits down. An observer records the total time taken for the whole TUG test using a stopwatch as a metric to explain the overall mobility [11,12,13]. As the TUG test contains various subtasks like sit-to-stand, walking, turning, and stand-to-sit, the robust and accurate elapsed time of each TUG subtask can provide important clinical information, such as elapsed time and speed of each TUG subtask, gait speed, cadence (number of steps per minute), and the stance (percentage of the gait cycle) of subjects [2,3,11,12,13,14,15]. These can be used not only to assist clinicians in clinical interventions but also to distinguish patients’ functional recovery [3,16]. More specifically, Salarian et al. [3] introduced the instrumented TUG (iTUG), which used portable inertial sensors to automatically detect and segment subtasks into two groups: early stages of PD versus age-matched control group. By comparing the two groups, they found significant differences between them in three subtasks of the iTUG, even though the total time had no significant difference in distinguishing the performance of the two groups. Furthermore, they showed that early PD subjects had unusual features, such as slow turning, arm swing, cadence, and trunk rotation, during straight walking [3]. Ansai et al. [16] also found that in the case of falls with mild cognitive impairment (MCI) and in non-fallers with Alzheimer’s disease (AD), there was a significant difference in turn-to-sit subtask performance, even though no other difference was found in the total consumption time between the two groups. Therefore, patients who could not be classified according to total time could be classified through subtask segmentation.

A typical TUG test is performed under a clinician’s supervision, which may require both physical and mental human efforts. Moreover, because the manual measurement using a stopwatch is a subjective observation, it could be inaccurate and inconvenient [1]. Furthermore, visiting a hospital to perform a TUG test is not trivial for patients who have limited mobility. This is a heavy burden for both patients and clinicians. Therefore, automatic TUG analysis methods, especially subtask segmentation methods [2,3,10,11,12,13,14,15,16], have recently attracted considerable research attention [4,17,18].

TUG tests have been widely studied not only for various patients but also in healthy young and older adults. For older adults, frailty is a common syndrome that embodies a high risk of critical declines, such as cognitive and functional decline, and falls with accelerated aging. Early diagnosis of a frailty syndrome can reduce harmful health outcomes and slow down the progression of frailty, for example, by prescribing suitable exercise programs [19,20,21]. This benefits individuals and relieves the burden on families and society. Healthy young people are also often selected as TUG subtask segmentation study subjects for several reasons: (1) to validate and evaluate the developed system [22,23,24,25]; (2) to adopt normal people as a criterion for comparison and perform an analysis on abnormal people (older adults, frailty syndrome, PD patients, and potential patients who do not yet show symptoms or those who already show symptoms) [24,25]; and (3) to obtain quick and meaningful data [25]. Therefore, this study collected data on both healthy young and older adults, and the performance of the proposed approach was then compared with those of the previous studies.

We think that there are two main challenges of TUG subtask segmentation studies: (1) accuracy improvement to provide more precise clinical information to patients and clinicians, and (2) automation to simplify TUG tests. If a TUG subtask is classified with higher accuracy, more precise clinical information, such as stride, gait speed, and turning speed, can be extracted. This may provide medical experts with a more elaborate diagnosis and therapy evaluation, e.g., after rehabilitation. In addition, automated TUG tests can reduce the test time and efforts of human operators, which may allow TUG tests at home for better accessibility to patients. For these benefits, the most advanced and readily available RGB-D cameras such as Azure Kinect must be used along with the current deep learning (DL) methods.

To the best of our knowledge, there are no previous studies on TUG subtask segmentation using a DL approach and an RGB-D camera. The RGB-D cameras have only been used in rule-based methods. The main contributions of this study are summarized as follows.

We propose a novel DL-based subtask segmentation of the TUG test using an RGB-D camera (Azure Kinect). In the proposed method, a dilated temporal convolutional network (TCN) is used to improve the accuracy and processing time of subtask segmentation compared to the existing Bi-LSTM.We investigated several inputs to the dilated TCN model to determine the input(s) that is (are) better than the others in the TUG subtask segmentation. We showed that the input from the pelvis alone had the best accuracy among many inputs. This single feature point and the optimized dilated TCN architecture also reduce the processing time in the inference phase.We evaluated the proposed method using the newly collected TUG data for three subject groups: (1) healthy young people, (2) older adults, and (3) stroke patients. The test results showed significantly better accuracy and robustness than existing rule-based and artificial neural network (ANN)-based subtask segmentation methods.

The rest of the paper is organized as follows: Section 2 introduces the related work on the automation of TUG. Section 3 provides the applied methodology with details. Section 4 contains experiments and results, and discussion is presented in Section 5. Finally, Section 6 concludes the paper and suggests some future work.

## 2. Related Work

The automation of the TUG test can be divided into contact and non-contact methods. In contact methods, wearable devices, such as inertial measurement units (IMU) or infrared sensors, are usually used. For example, Hsieh et al. [4] segmented subtasks for a TKA patient using inertial sensors with machine learning (ML) techniques as the classifier. Ortega-Bastidas et al. [26] used a rule-based technique to segment subtasks for 25 healthy young and 12 elderly subjects using inertial sensors and an RGB camera for ground truth data. In contrast, with older adults, Hellmers et al. [27] used only a single IMU and applied four ML methods: boosted decision trees (BDT), boosted decision stump (BDS), multilayer perceptron (MLP), and adaptive multihyperplane machine (AMM). They classified the subtask into static (sit, stand), dynamic (walk, turn), and transition (sit-to-stand, stand-to-sit) using BDT and classified subtask activities using MLP. For more details on contact methods, refer to [2,3,4,14,23,28,29]. Although these wearable devices generally show high resolution and accuracy, they require complex and inconvenient setup and calibration processes, increasing the physical burden on medical professionals and patients.

Non-contact methods primarily use non-contact sensors, such as RGB or RGB-D cameras, because video data can be collected from almost anywhere with simple settings. Using a single RGB camera along with deep learning (DL) and postprocessing techniques, Savoie et al. [22] segmented subtasks for healthy young people. First, 2D keypoints were extracted from the RGB image through the mask R-CNN [30], and 3D local poses were extracted using a deep multitask architecture for human sensing (DMHS) [31]. Then, these 2D keypoints and 3D local poses were used together to extract the global 3D poses of the participants. However, for segmentation, a rule-based technique was used, considering the characteristics of the trajectories of each joint. Li et al. [18] segmented subtasks for 24 patients with PD using an RGB camera and ANN-based technique. Although these methods generate good results, they typically require several steps to extract global 3D poses [22], or they allow only frontal camera locations [18].

In contrast with the RGB cameras, the RGB-D cameras, especially Kinect, are popular because of their accurate built-in skeleton tracking application programming interface (API) and silhouette extraction capabilities. Moreover, Kinect has proven sufficiently accurate compared to golden standards such as marker-based systems in gait [32,33,34]. Using RGB-D cameras, Lohmann et al. [35] segmented subtasks for normal older people using two Kinect Xboxes (RGB-D sensors). They used a rule-based technique with a specific feature, such as maximum or minimum shoulder *z*-axis acceleration, among the skeleton data. For other purposes, Jannis et al. [36] segmented phases (the same as subtasks) for patients with PD using Kinect V2. They also used a rule-based technique that exploited periodically increasing distances between the feet. The segmented phases were then used to assess the PD disease scale clinically using the TUG. Kampel et al. [37] proposed automatic TUG subtask methods for a functional decline assessment of 11 older adults using the Kinect V2 and rule-based technique with a specific feature, such as the velocity of shoulder *z*-axis, and with other features. Note that all these methods using RGB-D cameras were rule-based and not DL-based.

The TUG subtask segmentation approaches are primarily classified into two types: (1) rule-based and (2) ANN-based. The rule-based methods are applicable to a small number of datasets, as described in studies [3,23,28,35,37]. However, rule-based methods generally use complex predetermined rules to detect the motion transition following physical attributes, such as the trunk bending forward, and moving the feet [38]. For example, by using two Kinects to find local maxima, the skeleton TUG (sTUG) detected ten events: start moving, end uprising, start walking, start rotating, start turning, max turn, end turning, end rotating, start lowering, and end moving [37]. As another example, Ortega-Bastidas et al. [26] used a single wireless IMU sensor to detect turning start/end events by determining the average maximum/minimum *Yaw* (rotation) signal of an inertial sensor placed on the back. With their increasing complexity, analyzing and adjusting rule-based systems is often cumbersome. Therefore, these approaches require the careful selection of strict rules by experts, which may not be objective. Moreover, rule-based techniques may not be robust because a high variance in movement patterns can occur, especially for transitions, in older adults.

In contrast, the ANN-based subtask segmentation approach builds a model using features from training data with ground truth data generated by experts. Hsieh et al. [4] studied five ML techniques as classifiers: support vector machine (SVM), K-nearest neighbor, naïve Bayesian, decision tree, and adaptive boosting (Adaboost); data were obtained using six wearable sensors containing a triaxial accelerometer, a gyroscope, and a magnetometer. As another example, Li et al. [18] explored two pose estimators (interactive error feedback (IEF) and OpenPose) and two classifiers (SVM and bidirectional long short-term memory (Bi-LSTM)) using a single RGB camera. However, learning an ANN model requires a considerable number of TUG test datasets. Unfortunately, an open dataset related to the TUG test [17] cannot be used for TUG subtask segmentation because there is no ground-truth label for subtask segmentation.

## 3. Methods

### 3.1. Data Acquisition System and Participants

Figure 1 shows the system configuration for the 3 m TUG tests, which is similar to that in [37]. One difference is that the Azure Kinect was installed at a height of 1.5 m rather than 0.7 m. The *z*-axis (viewing direction) was perpendicular to the walking direction of participants for capturing left and right lateral motions, the *x*-axis was perpendicular to the walking direction, and the *y*-axis was for up and down movement. A cone was placed at 3 m straight from a standard chair with armrests to indicate a turning point where the participant should perform a “turn.” With this setting, participants sitting in a chair started to get up by the indicator’s “start” signal, walked 3 m at a comfortable walking speed, rotated the turning point (cone), walked 3 m back, leaned back on the chair, and then finally waited for the “end” signal. At this time, additional data were collected for one second before and after the “start” and “end” signals to prevent data from becoming unstable and ensure equal data lengths between the start and end periods upon data entry. The skeleton data were collected at a 30 Hz rate, and the RGB image, RGB-D image, and skeleton data were separately stored for subsequent analyses.

We used three groups of subjects: (1) 50 older adult subjects (15 male, 34 female, 79.37 ± 5.08 years old), (2) 15 healthy young subjects (13 male, 2 female, 28.75 ± 2.05 years old) that were confirmed to have no known walking disorders, and (3) 23 stroke subjects (18 male, 5 female, 53.89 ± 6.12). All experiments were conducted in various environments, such as apartments, laboratories, and fitness rooms, because of the subject’s availability in different locations. In this experiment, 15 trials of the 3 m TUG test for each participant were measured with a three-minute break after the first eight trials and one practice trial before the experiment. Some trials were excluded because of human error in measurement or missing skeletons. In this study, we obtained a TUG video dataset of 158 trials for healthy young subjects, 620 trials for older adults, and 268 trials for stroke patients. These datasets have an average video length of 11.34 s (healthy young), 15.33 s (older adults), and 43.11 s (stroke patients). The number of trials in this study is sufficient, considering a similar DL study with 24 PD patients and 127 trials using the Bi-LSTM model [18].

### 3.2. Proposed Method Overview

Figure 2 shows the overall process of the proposed method with typical time plots in each process. The raw trajectories (e.g., pelvis x, y, and z) are the normalized preprocessor inputs, and a low-pass filter (LPF) was used for the deep model training with the dilated TCN model. The predicted frame-level action states were postprocessed by dynamic time warping (DTW) to modify the misclassified action states. Finally, many action capabilities were computed for the total time, subtask times, subtask speeds, etc.

#### 3.2.1. Labeling Process

The proposed method detected six TUG events (StartMove, StartWalk, StartTurn, EndTurn, StartSit, and EndSit) and classified five subtasks (sit-to-stand, walk, turn, walkback, and stand-to-sit). The six TUG events were labeled for each TUG video to detect the six events. The labeling results of the two experts were averaged and used as the ground truth to avoid label biases by experts. We considered the labeling guidelines for six TUG events from [18], examples of which are summarized in Table 1. The reliability of the generated ground truth was measured using the intraclass correlation coefficient (ICC) [18]. The intra-rater reliability between the labeling results of the two experts showed high reliability with ICC = 0.99.

#### 3.2.2. Preprocessing

During preprocessing, an LPF was applied to remove noise generated during measurement, and min-max normalization was sequentially performed to normalize diverse ranges of subject data. Normalization of the trajectories is necessary because various height differences between subjects generate a diverse range of trajectories in the TUG data. In addition, to solve the problem of direction mismatch caused by camera coordinates, axis matching was performed to update the coordinates according to the subject’s walking direction [37]. Figure 3 shows an example of preprocessing for the pelvis trajectories. The optimal cutoff frequency selected by trial and error was 0.3 Hz, and the min-max normalization technique was finally utilized together by applying the 4th Butterworth LPF.

#### 3.2.3. DL-Based TUG Subtask Segmentation Algorithm

Action segmentation, such as the TUG subtask segmentation, is crucial to analyzing the activities of daily living. TUG subtask segmentation aims to segment all subtasks in time for a given TUG video. For frame-level action state classification in TUG subtask segmentation, we investigated a class of time-series DL models, called temporal convolutional networks (TCNs), because they can capture features of action durations and pairwise transitions between segments and long-range temporal patterns more efficiently using a hierarchy of temporal convolutional filters [39]. This model is faster to train through parallel operations based on dilated convolution and tends to outperform canonical recurrent architectures, such as LSTMs and gated recurrent units (GRUs), across a broad range of sequence modeling tasks [40].

The dilated TCN uses a deep series of dilated convolutions instead of pooling and up-sampling, adding skip connections between layers. Each layer uses dilated filters that operate on only a small number of time steps. The dilated TCN has the following properties: (1) computations are performed layer-wise, which means that every time step is updated simultaneously; (2) convolutions are computed according to the time; and (3) predictions at each frame are a function of the receptive field. Figure 4 shows the architecture applied to the proposed TUG subtask segmentation. The normalized and filtered trajectories were input to the first temporal block. Three temporal blocks generated features, and the softmax activation function in the final layer output the frame-level action state classification results.

#### 3.2.4. Postprocessing

The predicted frame-level action states from the dilated TCN can generate misclassified action states in each frame; this has already been discussed as a fragmentation error in [4,18]. Considering the strict time order of TUG subtasks (e.g., sit, sit-to-stand, walk, turn, walk-same as walk back, stand-to-sit, and sit), we also performed the DTW [18] algorithm to find the most appropriate subtask segmentation in the postprocessing stage. This was an application of the domain (field) knowledge in subtask classification to obtain the most appropriate subtask segmentation.

Figure 5 shows the example of the postprocessing for correcting the frame-level misclassifications. The green line represents the ground truth, the blue line represents the model classification result before postprocessing, and the red line represents the model classification result after DTW. In Figure 5a, the index of the *y*-axis represents each subtask, in which the walk and walk back subtasks existing between the turn subtasks are marked on the same index. Figure 5 shows the results before (Figure 5b) and after (Figure 5c) the DTW algorithm. Figure 5d shows the ground truth (green line) (Figure 5a) and the result (red line) (Figure 5c) from the DTW. Comparing Figure 5b,c illustrates that the prediction model incorrectly classified the “sit-to-stand” (index = 1) to “stand-to-sit” (index = 4) for the state transition from “sit-to-stand” to “walk” (index = 2). After the DTW was applied, this misclassification disappeared.

## 4. Experiments and Results

This section presents two studies for optimization of the input and DL models: (1) an input comparison study to find the most effective input for the proposed method, and (2) optimization of a DL model based on the results of the input comparison study. To demonstrate the superiority of the proposed method over the conventional methods, two methods were selected that are most similar to the proposed method regarding the experimental setup: (1) the rule-based TUG subtask segmentation method [37] and subject type (older adults); (2) the ANN-based TUG subtask segmentation method [22] and subject type (healthy young).

### 4.1. Metrics

The mean absolute error (MAE), standard deviation (STD), precision, recall, and F1 score were used to evaluate the TUG subtask segmentation classification performance for a fair and objective comparison with conventional methods. For the frame classification accuracy, we calculated the precision (prec.), recall (rec.), and the harmonic mean of the precision F1 score (F1) for each subtask, where TP, FP, and FN represent the number of true positive, false positive, and false negative frames for a given subtask. The total subtask classification accuracy (Acc.) was defined as the percentage of correctly predicted observations to the total number of observations in a trial. The equations below are for precision, recall, F1 score, and accuracy.
(1)$$\mathrm{Precision}=\frac{TP}{TP+FP}=\frac{TP}{All\;detections}$$
(2)$$\mathrm{Recall}=\frac{TP}{TP+FN}=\frac{TP}{All\;ground\;truths}$$
(3)$$F1\;\mathrm{score}=2\;*\;\frac{prec\;*\;rec}{prec+rec}$$
(4)$$\mathrm{Accuracy}=\frac{TP+TN}{TP+TN+FP+FN}$$

Recall evaluated cases in which the correct answer was inferred from the standpoint of the ground truth, and precision evaluated from the perspective of the classification model. The best result was when both the recall and precision were high. However, because these two are conflicting concepts, it is generally difficult to judge because it is a trade-off relationship that decreases when one increases. The F1 score was defined as the harmonic mean between precision and recall and was used as a statistical measure to rate performance. The total TUG time was defined as the elapsed time between the first (StartMove) and last (EndSit) TUG events. The total TUG MAE was the mean absolute difference between its time duration estimated by the model and the ground truth.

### 4.2. Input Comparison Study

Human activity can be detected using various sensors (inertial, RGB camera, and RGB-D camera (Kinect)) and classified by suitable DL-based methods. However, using raw images or video frames is usually very time-consuming, even in ANN-based approaches. Moreover, it may require considerable training data from diverse viewing angles and involves significant time and effort. Because the TUG test consists of relatively simple activities, it can be easily distinguished using low-dimensional features, such as human posture detected by body keypoints [18]. Therefore, previous studies investigated various types of input data to obtain efficient results with subtask segmentation [3,4,18,22,23,28,35,37,41] or fall risk prediction [25,42]. We summarize these in Table A1 in the Appendix A.

Figure 6 shows the total 32 skeleton joints detected using Azure Kinect. Among these, previous automatic TUG research with wearable sensors, such as a single IMU, argued that the proximal location of sensors (e.g., at the lower back) could more accurately detect the main gait events and spatiotemporal and kinematic factors [2,43]. Additionally, according to Sprint’s automatic TUG test research survey [2], 55.56% of wearable sensors, such as IMUs, are attached to the lower back. These findings motivated us to speculate that similar locations might be better than other locations in the TUG subtask segmentation task by using a lower resolution of RGB-D than the IMU. Previous studies using RGB or RGB-D cameras used complete skeleton data spread throughout the body as inputs to ANN models or rules [18,22,42]. We hypothesized that the input(s) only from the proximal joints, for example, center of gravity (COG) movement or head motion, and not with all 32 skeleton joints, should be sufficient for classifying TUG subtasks. Therefore, only five inputs (red box) were investigated for input comparison in this study because the accuracies of skeleton joint motion from the distal joints, such as ankles and hands, are usually very low due to fast movements and a wide range of motion. Using a smaller number of feature points can also significantly reduce the processing time in the inference phase, which is important for edge computing on mobile devices.

To determine which input(s) is (are) better than the others in TUG subtask segmentation, we investigated five different skeleton feature points from five groups of skeleton joints: (1) pelvis, (2) spine chest, (3) head, (4) a pair of left and right hands, and (5) a pair of left and right ankles (Figure 6). The pelvis and spine chest were selected because they are close to the COG inputs. The head, a pair of hands, and ankles were chosen to evaluate the effects of the inputs from the distal joints representing significant overall body motion.

Table 2 shows the results (5-fold average accuracy (%)) for the older adults, healthy young, and stroke patients with four different group combinations of skeleton feature points: (1) five single features (pelvis, spine chest, head, both hands, and both ankles), (2) four combinations of two features (pelvis + spine chest, pelvis + head, pelvis + both ankles, pelvis + both hands), (3) five combinations of three features (pelvis + spine chest + head, pelvis + head + both ankles, pelvis + both hands + both ankles, pelvis + head + both hands, and head + both hands + both ankles), and (4) a combination of all five features. Note that some feature combinations were excluded (e.g., spine chest + head, hands + ankles, head + spine chest + hands) because the distal joint inputs (hands and ankles) far from the COG have larger noise in the data as the accuracy was found to be worse. These results were obtained by training and testing each participant group separately.

Table 2 shows many interesting points: (1) a single feature of the pelvis had the highest accuracy of 95.46%, 94.53%, and 93.58% for healthy young, older adults, and stroke patients, respectively. This also shows that the accuracy decreased from the healthy young to older adults and stroke patients, as expected. In addition, the 5-fold average total TUG MAE (s) was 0.1449, 0.2213, and 0.5572 for the healthy young, older adult, and stroke patients, respectively, which indicates that the average total TUG MAE increases (almost doubled) from healthy young to older adult and stroke patients. (2) A single distal feature (head, hand, ankle) that is far from the COG showed relatively poor accuracy compared to the proximal features (pelvis, spine chest). This shows that the effects of the distal joint inputs are far from the COG. Note that the performance of the head input was better among these distal joint inputs (hand and ankle) because the head does not shake much, and skeleton tracking was easy. (3) The combination of more features on top of the pelvis showed no accuracy improvement for all subject groups. These results imply that the pelvis alone is sufficient for the subtask classification. Other previous approaches had also demonstrated the effectiveness of the pelvis point for other purposes for pathological gait classification with IMUs attached to the pelvis [43].

### 4.3. Optimization of Deep Learning Model

The proposed dilated TCN architecture, whose basic structure is shown in Figure 4, was optimized with pelvis input to determine the best kernel and window sizes. These two sizes are important hyperparameters in the dilated TCN architecture. We investigated three kernel sizes (3, 5, and 7) and four window sizes (4, 8, 16, and 32) because many other studies considered these ranges in the optimization process. Table 3 shows that a kernel size of three and a window size of eight showed the best performance for TUG subtask segmentation.

To find out the optimal hyperparameters, we used the grid search method: a heuristic approach. We considered the following hyperparameters: (1) learning rate in the interval of {0.00001, 0.00005, 0.00007, 0.0001, 0.0003, 0.0005}, (2) optimizer in the interval of {Adam, RMSprop}, (3) patience in the interval of {10, 20, 30, 50, 70, 100}, (4) batch size in the interval of {32, 128, 256, 512, 1024, 2048}. From this optimization process with the categorical cross-entropy loss in the proposed dilated TCN training, optimal parameters were determined as a learning rate of 7 × 10 − 5, optimizer of Adam, patience of 50, and batch size of 1024 to segment the TUG subtask (e.g., a multiclassification task). Figure 7 shows the behavior of loss and accuracy with the pelvis input. The best model was obtained using an early stopping criterion; the training will stop if the validation loss is not updated for a specific epoch (patience). Datasets were split into 6:2:2 (training: validation: test) to train the dilated TCN network for TUG subtask segmentation, e.g., (30: 10: 10) for 50 older adult datasets. To prevent overfitting, the model stopped training early when the validation loss did not improve during 50 epochs.

Based on the optimized parameters (learning rate of 7 × 10 − 5, optimizer of Adam, patience of 50, and batch size of 1024), we performed an ablation study to find out two optimal parameters of the dilated TCN model: (1) number of temporal blocks in the four ranges of {1, 2, 3, 4}, (2) number of convolutional layers in the three ranges of {1, 2, 3}. Five-fold cross-validation was used for the accurate evaluation and optimization of the model hyperparameters. The following results in Table 4 from the ablation study show the best accuracy for three temporal blocks with two convolutional layers for the TUG subtask segmentation task.

We also compared the performance between the dilated TCN and Bi-LSTM architecture in [18] for encoding the input features. Bi-LSTM can combine temporal information in the positive and negative time directions. It also contained separate layers for batch normalization and dropout. The encoded features were then sent to a fully connected layer using the softmax function to predict the frame labels. The optimized Bi-LSTM architecture had 128 layers and a dense layer of 64 neurons. The 5-fold accuracy of the proposed method with TCN with 41,879 parameters was 94.53%, whereas that of the Bi-LSTM with 546,181 parameters was 94.23%. This comparison shows that the proposed method had slightly better accuracy and a significantly smaller number of model parameters.

### 4.4. Comparison with Rule-Based Method

Considering the objectives of the experiments, the experimental setup of the RGB-D camera for data acquisition, and subjects (older adults), we compared the TUG subtask segmentation classification results of the proposed method with those of the rule-based skeleton TUG in [37]. The proposed approach showed the best performance among the skeleton TUG [37], depth TUG [37], and sTUG [35]. We compare the TUG event detection and TUG subtask segmentation in the following subsections.

#### 4.4.1. TUG Event Detection

To compare the results from the two methods, we first explain the TUG event detection (labeling) criteria because there are no standard criteria for segmenting subtask regions. Each subtask segmentation study uses its own criteria. The TUG event detection (labeling) criteria differ between the skeleton TUG and the proposed method. The skeleton TUG detected twelve events: the beginning and end of the six subtasks (ChairRise, FirstWalk, FirstTurn, SecondWalk, SecondTurn, SitDown). In this study, six TUG events (StartMove, StartWalk, StartTurn, EndTurn, StartSit, EndSit) were detected for the classification of five subtasks (sit-to-stand, walk, turn, walk back, and stand-to-sit). Note that the “walkback” of the proposed method can be represented with a combination of the “SecondWalk” and “SecondTurn” of the skeleton TUG. Furthermore, the “stand-to-sit” of the proposed method can be denoted as “SitDown” of skeleton TUG because “stand-to-sit” is the time elapsed from the moment the subject completes the turn around to sit back in the chair until the moment the subject’s back lies against the backrest of the chair. Other studies have also considered five subtasks, using the IMU in smartphones [44] or UWB radar and insoles sensors [45].

Comparing the six events in both the studies, only StartMove and EndSit showed apparent differences in detection. Our study defined the StartMove event as when the body was tilted by 45°, whereas the ChairRise start event of the skeleton TUG detected the time when the body started to tilt. Our study defined the EndSit event as when the body was tilted by approximately 45° to sit in a chair. The SitDown end event of the skeleton TUG detected the time when the body was fully stretched with a slope of less than zero. These criteria for the ChairRise start and SitDown end in the skeleton TUG were made automatically based on predefined rules. In contrast, in our study, StartMove and EndSit events were detected by two experts, and averages were used as final detection, similar to previous studies [18]. The differences in these event detection criteria for the two events turned out to be insignificant, as manifested in a later result section.

Despite different definitions of subtasks, six events had similar geometrical meanings in both studies, as shown in Figure 8: (1) StartMove = ChairRise Start, (2) EndSit = SitDown End, (3) StartWalk = FirstWalk Start, (4) StartTurn = FirstTurn Start, (5) EndTurn = FirstTurn End, and (6) StartSit = SecondTurn End. Therefore, these six events can be used to compare the event detection performance.

Because the TUG subtask was segmented based on TUG event detection, Figure 9 compares TUG event detection errors for older adults [37] (MAE and STD) for six events in the bar chart for a more transparent comparison between the two methods. The MAE and STD for TUG event detection of the two methods did not show a significant difference between StartMove and EndSit. This indicates that the difference in the detection criteria for these two events did not cause significant differences in the MAE and STD. However, the other intermediate events (StartWalk, StartTurn, EndTurn, and StartSit) showed significant differences. The results show that the proposed DL-based method detected TUG events far more accurately (smaller MAE) and robustly (smaller STD) than rule-based methods.

#### 4.4.2. TUG Subtask Segmentation

Table 5 shows the subtask segmentation comparison regarding MAE, precision, recall, and F1 score for older adults. Because the F1 score for each TUG subtask of the skeleton TUG was not computed in the skeleton TUG, the F1 score was calculated here for a fair comparison. Note that the results in the skeleton TUG did not include the STD of MAE, whereas those of the proposed method included the STD of MAE to show the variability of the MAE.

Table 5 shows that for all five subtasks, the proposed method showed significantly lower timing errors (MAE) (9–10 times) and higher F1 scores. This means that the proposed method had a better classification performance for the multiclass classification task with imbalanced data, such as the TUG test, in which data were collected in an unbalanced form because the execution time of each activity was different for each subtask. However, in the total TUG time, precision, and F1 score were slightly better in the skeleton TUG than in the proposed approach. This may be because the skeleton TUG had a clearer criterion for sit-to-stand and stand-to-sit by their rules. In contrast, the proposed TUG in this study used the subjective criterion (45-degree inclination angle from the TUG video by two experts).

Figure 10 shows the results in a bar chart for a straightforward comparison between the two methods. The STD is also computed in the case of the proposed approach to demonstrate its robustness, whereas the STD is not calculated in [37]. The robustness of the proposed method may also be far better than that of the rule-based approach (despite no STD) if we consider the robustness of the TUG event detection results in Figure 9.

### 4.5. Comparison with ANN-Based Method

We also compared the performance of the proposed method with an existing ANN-based vTUG technique [22]. This comparison should be fair in terms of 3 m TUG subtask segmentation on 30 healthy young people despite the differences in RGB installed in front of the subject [22] vs. the RGB-D installed at the subject’s side in our approach. Note that vTUG used an ANN-based method only to obtain a 3D global pose. However, for segmentation, a rule-based technique was used, based on the characteristics of the trajectories of each joint. Comparisons with other ANN-based methods [4,18] were not performed because of subject differences (PD in [18], TKA [4]).

Figure 11 compares TUG event detection errors (MAE and STD) for six events in a bar chart for a more explicit comparison between the two methods. The MAE and STD for TUG event detection of the two methods showed significant differences. Overall results show that the proposed DL-based method detected TUG events far more accurately (smaller MAE) and robustly (smaller STD) than the conventional ANN-based method [22]. A direct subtask segmentation comparison, however, cannot be conducted because vTUG only presented TUG events’ performance.

## 5. Discussion

We discussed many important points after presenting the results in Section 4; hence, we discuss only additional points here.

In some cases, DTW cannot correct other types of frame-level misclassifications, such as the ambiguity (or shift) error [4] caused by an ambiguity between successive subtasks, such as sit-to-stand, walk/turn, turn, and walk. In this case, the effect of DTW is small, resulting in a minor performance improvement. However, as shown in Figure 5, in most cases, it works well for frame-level misclassification.

The applied domain knowledge-based postprocessing step is a usual process in many ML/DL methods for action classification, such as subtask segmentation to obtain better accuracy, e.g., [4,18]. Unfortunately, fair comparisons with other ANN-based methods [4,18] cannot be performed because of subject differences (PD in [18], TKA [4]). Additionally, one should note that an existing ANN-based vTUG method [22] used an ANN only to obtain a 3D global pose. However, for segmentation, a rule-based technique was used, considering the characteristics of the trajectories of each joint. Therefore, we cannot compare our performance by using DTW with this method.

Furthermore, to fairly compare the performances using RGB-D data with those using RGB data in the proposed approach, the pelvis point (x,y,z coordinates) must be captured in the RGB images. However, the pelvis point (x,y,z coordinates) must be captured in the RGB images to compare the performance of RGB-D data with RGB data in the proposed approach. However, this requires a physical marker on the subject, the location of which may also be different from that of the pelvis point that is captured by an RGB-D camera. As an alternative to the pelvis point, the COM (Center of Mass) may be extracted from RGB data. However, the COM is different from the pelvis point, which hinders a fair comparison between the performance with RGB-D data with that of RGB data. Moreover, COM computation requires many processes such as silhouette/background segmentation, background elimination, and COM computation from silhouette, which may accumulate errors in each process. Most seriously, the computed COM may not be robust because of pixel-based COM computation, as previously found in [22]. Although this study presented the results of the subtask segmentation for stroke patients, these results cannot be compared with other methods because of the unavailability of results for stroke patients from different approaches that use the same RGB-D cameras.

## 6. Conclusions

A novel DL-based subtask segmentation method was proposed for TUG tests using a single RGB-D camera and a dilated TCN. An evaluation of newly collected TUG data for three different subject groups (healthy young, elderly adults, and stroke patients) showed that the proposed method is more robust and accurate than the rule-based and ANN-based subtask segmentation methods. The evaluation results (healthy young = 95.458%, healthy adult = 94.525%, and stroke patients = 93.578%) demonstrated the generality and robustness of the proposed method. Moreover, an investigation of several inputs to the dilated TCN model showed that the input only from the pelvis is enough to achieve the best accuracy among many inputs. In addition, this single feature point significantly reduces the DL model’s memory requirement and inference phase’s processing time. If the TUG subtask is classified with higher accuracy, more precise clinical data (e.g., stride, gait speed, turning speed, etc.) can be extracted, which may provide medical experts with more elaborate diagnosis and therapy evaluation, e.g., for rehabilitation.

In the future, we will apply the proposed method to more diverse patient groups such as PD, MCI, AD, TKA, MS, lumbar degenerative disc disease, lower limb amputations, and COPD. In addition, we will implement the trained neural network model in the edge devices such as smart mobile phones for better portability. Finally, we will develop a method to correct shift errors for better accuracy.

## Figures and Tables

**Figure 1 sensors-22-06323-f001:**
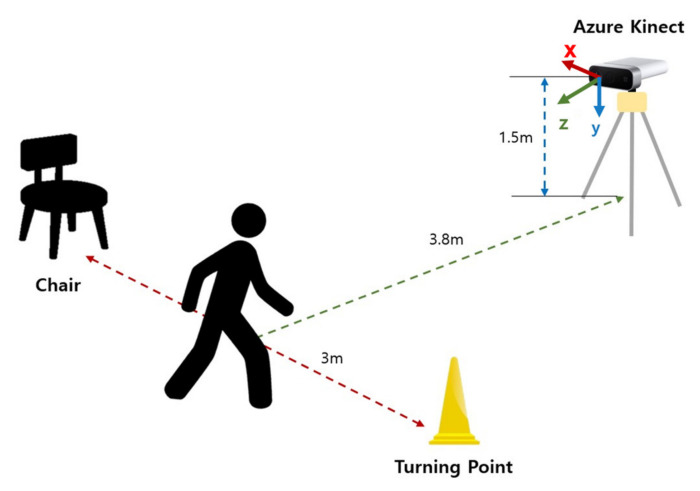
System configuration for the study. The participants performed a 3 m TUG test. A cone was placed at 3 m straight from a standard chair, and Azure Kinect was installed perpendicular to the walking direction at the height of 1.5 m.

**Figure 2 sensors-22-06323-f002:**
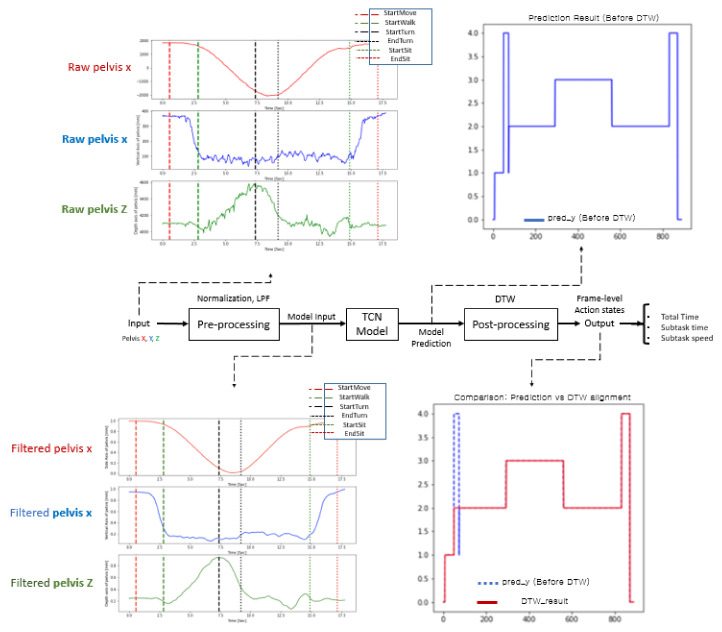
Overall flowchart of the proposed method.

**Figure 3 sensors-22-06323-f003:**
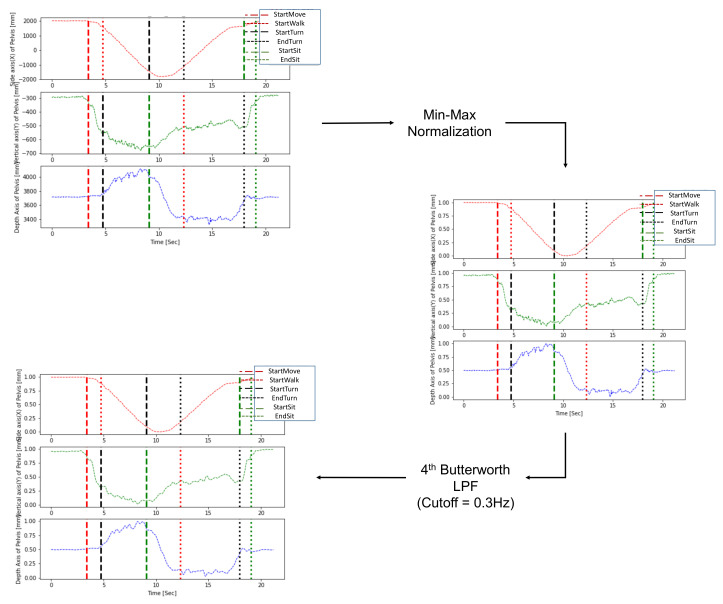
Preprocessing by low pass filter and normalization for a pelvis trajectory.

**Figure 4 sensors-22-06323-f004:**
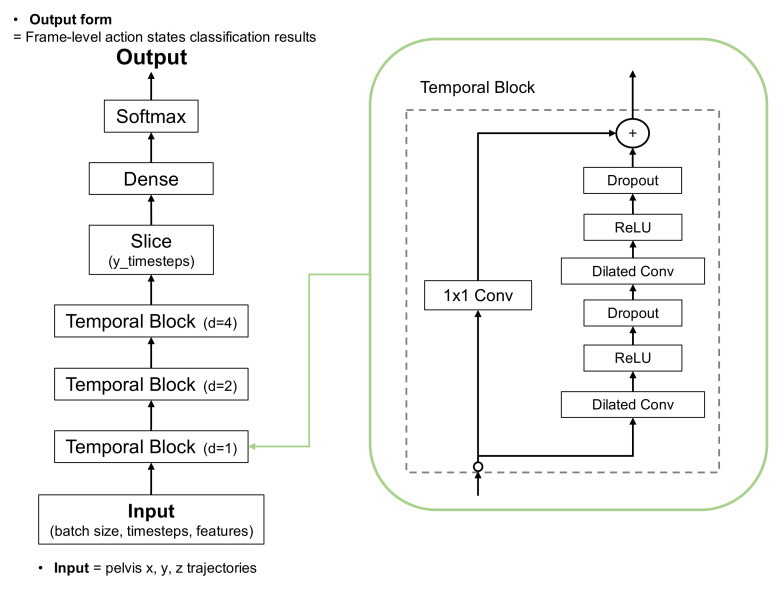
Architecture of dilated temporal convolutional network (TCN).

**Figure 5 sensors-22-06323-f005:**
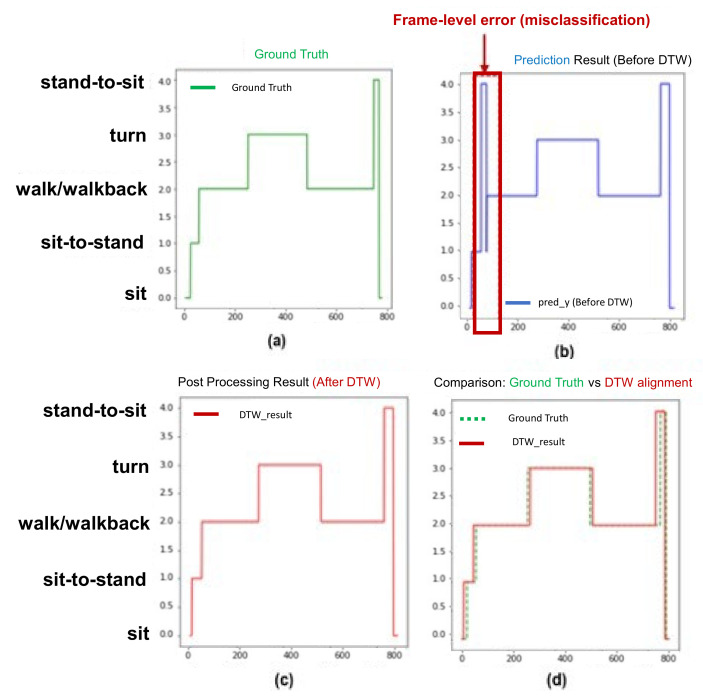
Sample postprocessing for correcting frame-level misclassification.

**Figure 6 sensors-22-06323-f006:**
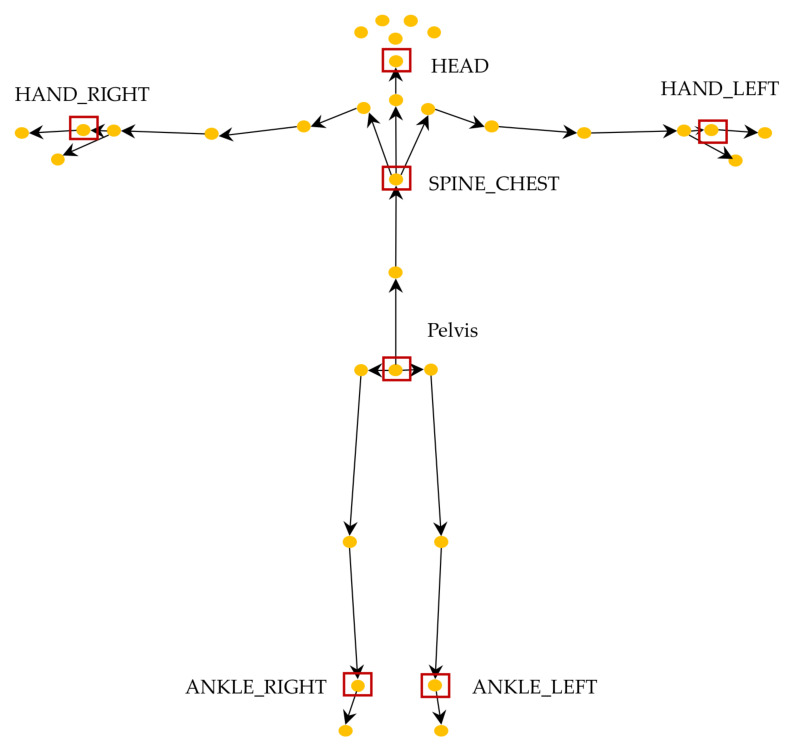
Skeleton joints tracked by the Azure Kinect and used inputs for comparison (red box).

**Figure 7 sensors-22-06323-f007:**
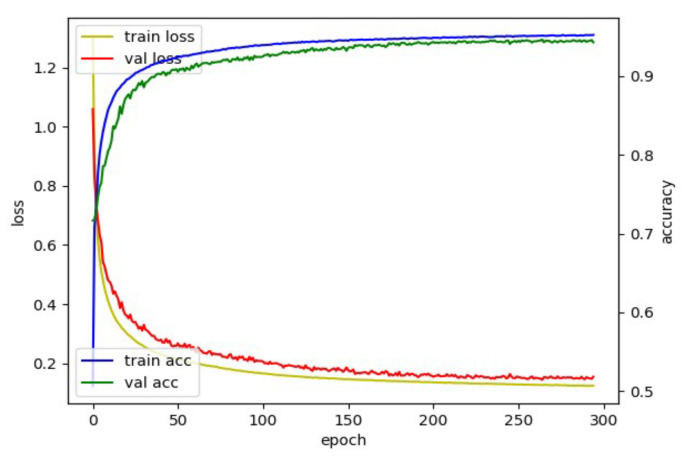
Loss and accuracy plot for model with pelvis input.

**Figure 8 sensors-22-06323-f008:**
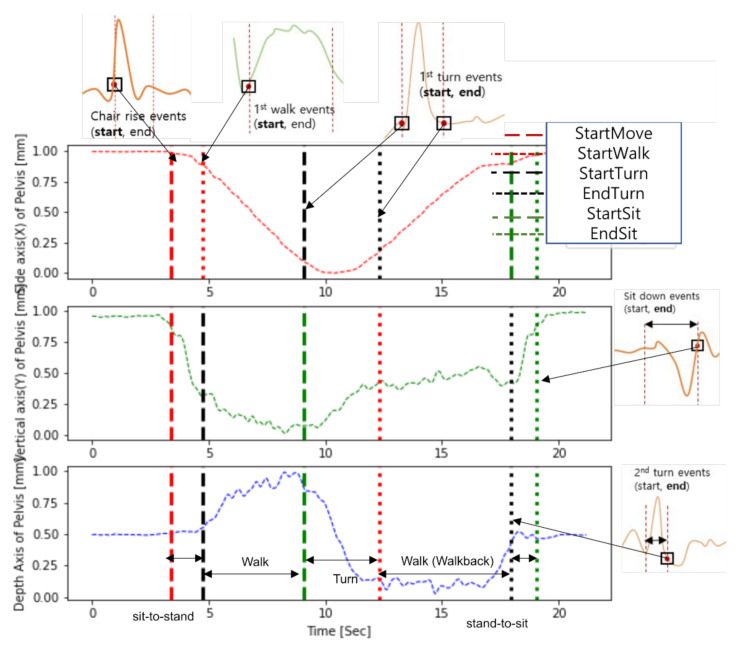
TUG events in Skeleton TUG and this study. Based on the dataset obtained in the study, it shows how the proposed method and the events of the skeleton TUG are matched, re-spectively. Each truncated plot is the result of detection of the event of the skeleton TUG.

**Figure 9 sensors-22-06323-f009:**
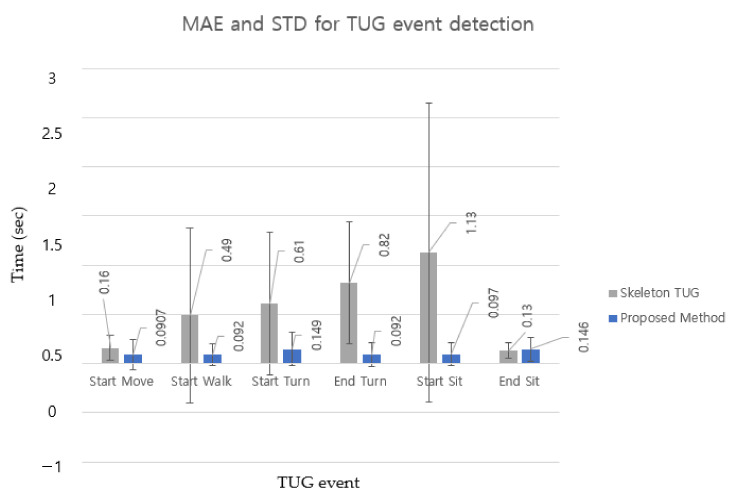
MAE and STD in seconds between skeleton TUG (gray bar) and the proposed method (blue bar) for each TUG event. Error bars are ± the STD of the values.

**Figure 10 sensors-22-06323-f010:**
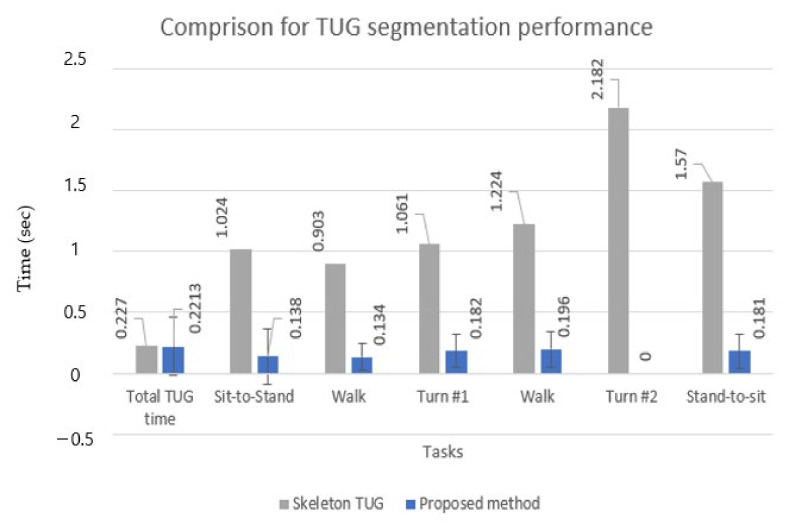
Comparison of MAE and STD for total TUG time and subtask segmentation.

**Figure 11 sensors-22-06323-f011:**
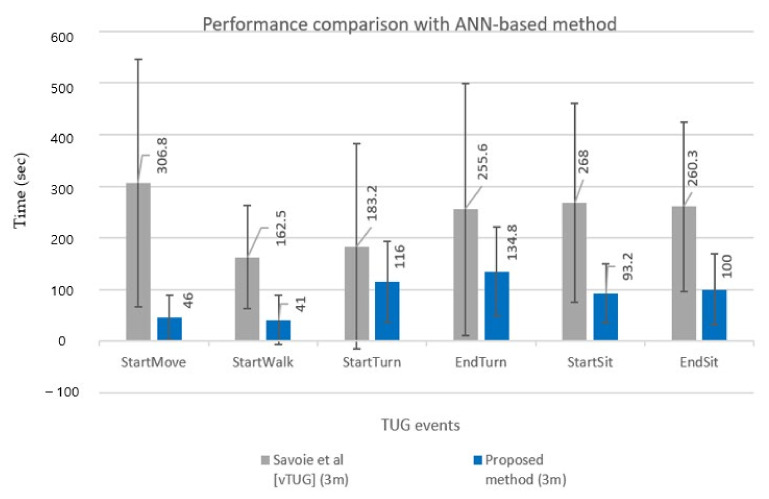
Comparison results with ANN-based method [22].

**Table 1 sensors-22-06323-t001:** Labeling guidelines for six events.

TUG Events	Label	Criteria
StartMove	0	When body is tilted 45 degrees to get up from the chair
StartWalk	1	After getting up from the chair, when the first step is off the ground
StartTurn	2	When subject rotates the body to turn at the TUG marker
EndTurn	3	After turning at the TUG marker, when the body looks back at the chair
StartSit	4	When body stands against the chair after turning body to sit on the chair
EndSit	5	When body is titled 45 degrees to lean on the chair

**Table 2 sensors-22-06323-t002:** Comparison results of TUG subtask segmentation accuracy of joints closest to COG.

	Input		Healthy Young	Older Adults	Stroke Patients
	Joint	No.	Acc. [%]	Acc. [%]	Acc. [%]
Results	pelvis	input 1	95.46	94.53	93.58
	spine chest	input 2	94.29	94.25	92.83
	head	input 3	94.1	93.86	90.86
	hand (left/right)	input 4	92.24	91.32	79.46
	ankle(left/right)	input 5	89.89	86.53	80.58
	pelvis, head	input 6	94.4	94.22	91.59
	pelvis, spine chest	input 7	94.46	94.045	92.29
	pelvis, ankle	input 8	93.42	93.81	87.56
	pelvis, hand	input 9	93.46	93.68	87.07
	pelvis, head, spine chest	input 10	93.78	93.44	91.72
	pelvis, head, ankle	input 11	93.59	93.93	90.89
	pelvis, hand, ankle	input 12	93.15	93.84	91.944
	pelvis, head, hand	input 13	93.3	93.63	91.31
	head, hand, ankle	input 14	93.25	93.39	91.72
	pelvis, head spine chest, hand, ankle	input 15	93.62	93.95	92.4

**Table 3 sensors-22-06323-t003:** Optimal values of kernel and window sizes.

Title 1	Kernel Size (Window Size = 8)			Window Size (Kernel Size = 3)			
	3	5	7	4	8	16	32
Accuracy	94.53	93.21	92.73	92.8	94.53	92.11	87.26
# of parameters	41,879	112,537	218,521	40,921	41,879	43,801	47,641

**Table 4 sensors-22-06323-t004:** Accuracy from ablation study of dilated TCN.

	Number of Temporal Blocks. Number of Conv. Layers		
	1.1	1.2	1.3
Acc [%]	92.3	93.53	92.54
	2.1	2.2	2.3
Acc [%]	93.94	90.42	92.03
	3.1	3.2	3.3
Acc [%]	92.34	94.53	93.44
	4.1	4.2	4.3
Acc [%]	92.75	92.31	91.86

**Table 5 sensors-22-06323-t005:** Comparison for TUG subtask segmentation for older adults.

Method	MAE, STD, Precision, Recall, and F1 Score of TUG Phases							
	Metric	Total TUG Time	Sit-to-Stand	Walk	Turn #1	Walk	Turn #2	Stand-to-Sit
Skeleton TUG [37]	MAE	0.227	1.024	0.903	1.061	1.224	2.182	1.570
	Prec.	0.997	0.647	0.961	0.793	0.831	0.832	0.593
	Recall	0.990	0.928	0.906	0.871	0.983	0.759	0.952
	F1 score	0.994	0.753	0.933	0.830	0.900	0.793	0.731
Proposed method	MAE	0.221	0.138	0.134	0.182	0.196	-	0.181
	STD	0.237	0.228	0.109	0.136	0.148	-	0.145
	Prec.	0.986	0.955	0.947	0.967	0.913	-	0.884
	Recall	0.990	0.973	0.966	0.96	0.932	-	0.818
	F1 score	0.988	0.964	0.957	0.963	0.923	-	0.849

## Data Availability

Data sharing is not applicable to this article.

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
