# Peer review of "Deep Learning-Based Subtask Segmentation of Timed Up-and-Go Test Using RGB-D Cameras"

_sensors, 2022, doi:10.3390/s22176323_

Round 1

Reviewer 1 Report

This paper considers the subtask segmentation of TUG test for the purpose of evaluating an individual’s basic functional mobility using RGB-D data. The topic is interesting and of practical importance. A deep learning based method is designed to segment subtasks via classifying individual frames. The method is clearly described, and abundant experimental results are provided. However, the following two important issues need to be clarified. 

1. This paper used DTW as a postprocessing to compare the prediction results with ground-truth. In standard machine learning paradiagm, ground-truth during test is used only to evaluate the performance and cannot be used to improve final prediction, otherwise the comparison with other methods is unfair. Moreover, the ground-truth is often unavailable during practical usage. 

2. This paper uses RGB-D data, and the compared methods used RGB data. So, how about comparison between these methods with only RGB data? This is necessary to show the advantages of the proposed method that are not endowed by the depth. Moreover, ablation study is needed to show the importance of depth information alone. 

Minor points: some legends in figs.2 and 3 are not clear.

Reviewer 2 Report

The authors conducted the study with accuracy and competence. The results are interesting.

Author Response

No replies are necessary for Reviewer 2 because there are no specific comments.

Reviewer 3 Report

1. The abstract should be revised to include the concise details of the key results of the proposed model. Add brief information about the quantitative measures and the impact of this work in comparison with other similar works.

2. The introduction section - Include a paragraph explaining the current challenges of TUG subtask segmentation studies using RGB-D cameras. 

3. The main contributions of this study could be moved to the position before the last paragraph of the introduction section.

4. The figure 1 caption fails to suitably address the contents of this diagram. Consider modifying it. Also, modify the caption of figure 9.

5. It is suggested to improve the presentation of the dataset description used in this work. Suggestion to include a separate sub-section for the presenting the holistic details of the dataset.

6. Improve the visual representation, resolution and readability of the figures 2, 3 , 5 and 8. The textual information of the graphs/images in these figure are not clear. Kindly consider including vector graphics images for all the figures. 

7. The equations in section 4.1, should be numbered. Kindly improve the presentation of these equations and their numbering.

8. In Section  4.3, discuss how the hyper-parameters were chosen for the proposed model. Is it heuristic based or general criterion irrespective of the training dataset? Authors need to provide justifications for all the hyper-parameters settings. How did you determine the values of these parameters? Are these hyper-parameter values optimal. Kindly consider including a table for all the hyper-parameter settings of the proposed model.

9. For performance evaluation, the data distributions for training, testing and validation have to be clearly pointed out.  The validation of the results of the proposed model is not clear. Provide more information about how all the outcomes of the proposed model were validated.

10. Include an ablation study of the proposed method.

11. In the conclusion section, the authors state the following " The proposed method can provide more accurate clinical information to medical experts in a more practical and efficient manner". Kindly provide the justification for this statement. Consider improving the presentation of the future works. 

12. The paper has some flaws in English language presentation. It requires English language editing by a native speaker.

Round 2

Reviewer 1 Report

Thanks for the clarification. I recommend to accept in present form.

Reviewer 3 Report

Most of the reviewers suggestions and comments are successfully addressed by the authors.